# Impact of an Acquisition Advanced Practice Provider on Home Hospital Patient Volumes and Length of Stay

**DOI:** 10.3390/healthcare11030282

**Published:** 2023-01-17

**Authors:** Heidi M. Felix, Jed C. Cowdell, Margaret R. Paulson, Karla C. Maita, Sagar B. Dugani, Francisco R. Avila, Ricardo A. Torres-Guzman, Antonio J. Forte, Gautam V. Matcha, Michael J. Maniaci

**Affiliations:** 1Division of Hospital Internal Medicine, Mayo Clinic, 4500 San Pablo Rd, Jacksonville, FL 32224, USA; 2Division of Hospital Internal Medicine, Mayo Clinic Health Systems, 1221 Whipple St, Eau Claire, WI 54701, USA; 3Division of Plastic Surgery, Mayo Clinic, 4500 San Pablo Rd, Jacksonville, FL 32224, USA; 4Division of Hospital Internal Medicine, Mayo Clinic, 200 1st St SW, Rochester, MN 55905, USA

**Keywords:** hospital-at-home, home healthcare, length of stay, advanced practice providers

## Abstract

In July 2020, Mayo Clinic introduced a hospital-at-home program, known as Advanced Care at Home (ACH) as an alternate option for clinically stable medical patients requiring hospital-level care. This retrospective cohort study evaluates the impact of the addition of a dedicated ACH patient acquisition Advanced Practice Provider (APP) on average length of stay (ALOS) and the number of patients admitted into the program between in Florida and Wisconsin between 6 July 2020 and 31 January 2022. Patient volumes and ALOS of 755 patients were analyzed between the two sites both before and after a dedicated acquisition APP was added to the Florida site on 1 June 2021. The addition of a dedicated acquisition APP did not affect the length of time a patient was in the emergency department or hospital ward prior to ACH transition (2.91 days [Florida] vs. 2.59 days [Wisconsin], *p* = 0.22), the transition time between initiation of the ACH consult to patient transfer home (0.85 days [Florida] vs. 1.16 days [Wisconsin], *p* = 0.28), or the total ALOS (6.63 days [Florida] vs. 6.34 days [Wisconsin], *p* = 0.47). The average number of patients acquired monthly was significantly increased in Florida (38.3 patients per month) compared with Wisconsin (21.6 patients per month) (*p* < 0.01). The addition of a dedicated patient acquisition APP resulted in significantly higher patient volumes but did not affect transition time or ALOS. Other hospital-at-home programs may consider the addition of an acquisition APP to maximize patient volumes.

## 1. Introduction

The average number of days patients spend in the hospital is known as the average length of stay (ALOS), and this represents a feasible indicator of hospital efficiency [1,2]. Several factors affect the ALOS, including the patient’s demographic characteristics, diagnosis, comorbidities, adverse events, and infections acquired during hospitalization [3,4,5]. Prolonged ALOS has been linked to negative patient outcomes, including increased likelihood of contracting a hospital acquired infection, suffering a venous thromboembolism, or experiencing cognitive impairment and decreased functional status [6,7,8]. Decreasing the ALOS may prevent these negative outcomes as well as help reduce healthcare expenses [9]. These savings may be attributed to a decrease in bed occupancy, medical errors, hospital-acquired conditions, and readmissions [10].

Telemedicine implementation has led to a significant reduction in hospital readmission and ALOS [11]. The expansion of patient management in outpatient settings using telemedicine has increased hospital bed availability and decreased hospital costs [12]. Both patients and providers have reported a positive experience with telemedicine [13,14,15], and isolated patients without the means to consistent healthcare facility access are now able to connect to medical providers in a whole new way [16]. In 2020, Mayo Clinic launched Advanced Care at Home (ACH) as a hybrid healthcare system that utilizes telemedicine and technological advances, allowing real-time video–audio connections, remote patient monitoring, and rapid response service to deliver hospital-level care in the comfort of the patient’s home.

Advanced practice providers (APPs), comprised of nurse practitioners and physician assistants, are vital to the ACH multidisciplinary team. The APPs primary role in ACH is to manage the day-to-day patient care activities, either by conducting virtual patient assessments from the ACH command center, or by traveling to patient homes to conduct a hands-on assessment. They are responsible for most of the ACH clinical documentation and orders. In addition to these clinical duties, APPs also review a list of ACH-eligible patients in the Emergency Department (ED) or hospital wards and transfer these patients to ACH. If the APPs clinical burden is high, patient acquisition is often delayed in order to complete vital patient care activities. Delays in patient acquisition may lead to lower patient volumes and higher ALOS. Hence, we hypothesized that adding an APP dedicated solely to ACH patient acquisition will both increase the volume of patients admitted to the ACH program as well as expedite patient transition from the ED or hospital wards into ACH, resulting in an overall lower hospital ALOS.

## 2. Materials and Methods

### 2.1. Patient Population and Clinical Setting

The study was approved by the Mayo Clinic Institutional Review Board (protocol number 20-010753). The study was conducted between 6 July 2020 and 31 January 2022, at Mayo Clinic Hospital in Florida, a 306-bed community academic hospital in Jacksonville, Florida, and Mayo Clinic Health System Eau Claire Hospital, a 304-bed community hospital in Eau Claire, Wisconsin. Mayo Clinic Florida is located in the urban environment of Duval County, Florida, with US Census Bureau data from the 2020 census reporting a population density of 1231 persons per square mile. Mayo Clinic Health Systems Hospital in Eau Claire, WI, is located in Eau Claire County, a very rural environment with a population density of 165 persons per square mile. Patient acquisition was done without dedicated acquisition personnel at the Florida and Wisconsin sites from 6 July 2020, through 31 May 2021. A dedicated ACH acquisition APP was added to the Florida site on June 1, 2021, and continued through the end of the data collection on 31 January 2022. Admission to the ACH program is voluntary, and patients provided oral and written consent to participate in the ACH program. The inclusion criteria were: presence of a clinical diagnosis supported by ACH, patient geography, payer source that supported ACH care, and home safety. Patients were excluded if they did not want to participate in the ACH program, met ACH exclusion criteria (uncontrolled mental illness, intravenous pain medication need, immobility or deconditioning that required 2-person 24-hour assistance, home environment not conductive to participation, and unstable arrhythmias requiring continuous telemetry), or had missing or unknown data.

### 2.2. The ACH Model of Care and the Acquisition APP Role

ACH integrates medical care directed by physicians, APPs, and registered nurses (RNs) located in a remote command center (CC) with in-home care executed by a medical supply chain located in close proximity to the patient [17]. Only patients residing in an inpatient designated space, either an Emergency Department or inpatient hospital ward, qualify for transfer into ACH. Patients requiring inpatient-level care in one of these two settings are screened for clinical stability, demographic eligibility, and home safety. If they qualify and consent to participation in the program, patients are transported via medical transport to their home where in-home technology is set up for high-acuity home care. Patients’ vital signs and clinical condition are monitored by combining Bluetooth-enabled biometric monitoring devices with a specially configured audio/video tablet that is directly connected to the clinical team in the CC. This technology transmits biometric data, such as blood pressure, heart rate, and oxygen saturation, to the CC where it is monitored by virtual-bedside RNs. Physicians and APPs round on the patients virtually each day while in-person assessments are done by a combination of a traveling RN, an APP, and/or a community paramedic at least twice daily. Additionally, the CC RNs conduct virtual assessments every 3–6 h and as needed. Based on the incoming data, the CC physician and nurse adjust the care plan. To execute in-home care delivery quickly to patients in ACH, an extensive vendor-mediated supply chain was created in each geographical area where ACH operates. This includes in-home rapid response services, phlebotomy, medication administration, nursing care, meals, and diagnostic images such as abdominal and chest radiographs and ultrasounds. After primary diagnosis stability is reached in the home setting, the patient is discharged from the ACH, the equivalent to being discharged from a brick-and-mortar (BAM) hospital.

Since the inception of ACH, the recruitment of patients to the ACH program has relied on ACH team members finding time between clinical duties to conduct one of four key duties. First, a clinical provider reviews the ED and hospital ward list which has already been filtered for ACH eligibility by geographical location and correct insurance provider, in order to determine which patients have the correct clinical diagnosis that could be treated in the program. These patients are deemed “possible admissions” for the program and receive an ACH consult order/designation. Separately, ED and inpatient providers can place an ACH consult order in the electronic health record (EHR) to initiate an ACH provider review for patient ACH eligibility. Second, chart reviews of screened patients are conducted to ensure that no surgeries, invasive procedures, or complex imaging studies are planned in the next 48 h. Third, patients are interviewed to ensure that they understand the program, express interest in participating, pass a social stability screen for in-home care, and provide informed consent to participate. Finally, consented patients must have a physical examination by the ACH provider, along with a documented transfer note and corresponding transfer orders. Prior to the addition of a dedicated acquisition APP, this four-step recruitment process is conducted by multiple providers at various times of the day, often with the overnight CC RNs screening for eligible patients, the traveling field APP reviewing the charts for clinical stability prior to embarking on in-home patient rounds, a case manager or available provider meeting with the patient and gauge program interest, the daytime RNs conducting the social stability screening by phone, and any available clinical provider conducting the exam and write the note and orders.

In May 2021, an acquisition APP was instituted at the Florida site to optimize provider usage and maximize patient recruitment. A patient registry was built in the EHR to filter patients eligible for ACH by geographical location and payer source in real time. The dedicated acquisition APP would have access to this registry and be housed in the BAM. Beginning daily at 7am, they could review the patients for clinical stability, travel to the ED or inpatient wards to meet with patients, and complete the four processes listed above for recruitment and transfer. The workflow was reviewed and approved by all ACH team members and several simulation exercises were conducted to ensure that the APPs dedicated to ACH patient acquisition understood the process. On 1 June 2021, the Florida site started to recruit all patients to ACH through the dedicated acquisition APP. The Wisconsin site continued to recruit patients through the traditional multi-provider method.

### 2.3. Data Collection and Statistical Analysis

Patient data were extracted from the EPIC electronic health record system (Epic Systems, Verona, WI, USA). These data were exported to an encrypted Microsoft Excel 2008 (Microsoft Corporation, Redmond, WA, USA) spreadsheet. Demographic data including patient age, sex, race, and ethnicity were collected. Study data collected included the length of time patients stayed in the ED or on the hospital ward prior to transferring into ACH, the length of time the patients were in the ACH virtual inpatient care program, the transition time between the placing of an ACH consult and the placement of the APP “transfer to ACH” order, and the total number of patients admitted to the program.

All ACH patients enrolled over the study timeframe were analyzed. A post hoc statistical power analysis was conducted to test between two independent groups using a two-tailed t-test, a medium effect size (d = 0.5), and p-size of 0.05 [18]. After data extraction from the EHR, patient age was analyzed with the average and standard deviation (SD) measurements performed using SPSS Statistics (IBM, Armonk, NY, USA). Patient volume data and ALOS data were analyzed using a two-sample *t*-test. A Pearson’s Chi-squared test was used to analyze patient sex and the Fisher’s Exact test was used to analyze patient race and ethnicity. A *p*-value of ≤0.05 was considered statistically significant.

## 3. Results

Between 6 July 2020 and 31 January 2022, a total of 753 patients were admitted to the ACH program, with 455 (60.4%) being admitted at the Florida site and 298 (39.6%) at the Wisconsin site. Two hundred and sixty-nine patients (35.7%) were admitted prior to the addition of the acquisition APP on 1 June 2021 (149 patients in Florida, 124 patients in Wisconsin), and 484 patients were admitted after 1 June 2021 (306 patient in Florida, 174 in Wisconsin). Post hoc statistical power analysis showed that a total post-intervention patient sample size of 484 patients achieved a power of 1.00 [18]. The mean patient age was 70.2 years (SD ± 15.1), 411 patients (54.6%) were male, 686 patients (91.1%) were of white race, and 39 patients (5.2%) were of Hispanic or Latino ethnicity, although when comparing the two geographical locations, the Florida site had a significantly more diverse race and ethnicity profile (Table 1).

Prior to the addition of the acquisition APP (6 July 2020–31 May 2021) there was no significant difference between the two geographical sites in terms of ED/hospital wards LOS (2.40 days [Florida] vs. 2.00 days [Wisconsin], *p* = 0.17), ACH virtual inpatient care LOS (3.72 days [Florida] vs. 3.87 days [Wisconsin], *p* = 0.65), the combined physical and virtual hospital LOS (6.12 days [Florida] vs. 5.87 days [Wisconsin], *p* = 0.64), the transition time from ACH consult to home admission (1.32 days [Florida] vs. 1.87 days [Wisconsin], *p* = 0.10), and the average number of patients acquired by ACH monthly (13.6 [Florida] vs. 12.4 [Wisconsin], *p* = 0.67). After the introduction of the acquisition APP at the Florida site, comparison of the two geographical sites revealed no significant difference between the two geographical sites in terms of ED/hospital wards LOS (2.91 days [Florida] vs. 2.59 days [Wisconsin], *p* = 0.22), ACH virtual inpatient care LOS (3.71 days [Florida] vs. 3.75 days [Wisconsin], *p* = 0.88), the combined physical and virtual hospital LOS (6.63 days [Florida] vs. 6.34 days [Wisconsin], *p* = 0.47), or the transition time from ACH consult to home admission (0.85 days [Florida] vs. 1.16 days [Wisconsin], *p* = 0.28). There was a significant increase in the average number of patients acquired by ACH monthly (38.3 in Florida vs. 21.6 in Wisconsin, *p* < 0.01) (Table 2).

A second analysis reviewing each site’s data separately before and after the introduction of the acquisition APP in Florida was conducted to account for program-specific changes (ACH program changes implemented at both sites including changes in staffing, automation, software, improved transportation resources, and improved ACH workflows) that may have improved LOS or patient volumes at both sites independent of the addition of the acquisition APP. Independently at each site, there was no significant difference in terms of ACH virtual inpatient care LOS, the combined physical and virtual hospital LOS, or the transition time from ACH consult to home admission. The Florida site had no significant change in ED/hospital wards LOS (2.41 days prior vs. 2.91 days post, *p* = 0.13) while Wisconsin did have a significant increase in ED/hospital wards LOS (2.00 days prior vs. 2.59 days post, *p* = 0.01). Both sites had significant increase in patient volumes, with Florida going from 13.6 monthly admissions to 38.3 (*p* < 0.01) and Wisconsin going from 12.4 to 21.8 (*p* < 0.01). This indicates that the Florida site had a 182% increase in volume after the addition of the acquisition APP while Wisconsin had an 75.5% increase in volume over the same period with no intervention (Table 3).

## 4. Discussion

We found that adding a dedicated acquisition APP to our ACH program was successful in significantly increasing program admission volumes but did not significantly affect our ALOS or transition time from BAM to home. These findings are important for several reasons. First, to make virtual hybrid HaH programs affordable and attractive to both commercial payers and health systems, significant scaling of patient volumes must occur. This is because this model relies on vendor-mediated resources in the community acting as a high-acuity medical supply chain. Maintaining these high-cost resources at the ready but minimal patient volumes receiving their care would not be cost-effective. Having a balanced and adequate patient volume in the ACH model to match these resources is essential. We found the acquisition APP provided a 182% increase in patient volumes which was significantly more than the gradual 75.5% increase seen at our control site (Figure 1). This type of rapid and expansive recruitment of appropriate patients is necessary to the success of HaH. Furthermore, hospitals dealing with over capacity situations need a way of quickly getting patients into alternative care settings. The acquisition APP met these needs. In addition, we found that our Florida campus had a more racially and ethnically diverse patient population (13.2% and 6.4% respectively) when compared to our Wisconsin campus (1.3% and 3.4%). This is likely due to the urban nature of the Florida campus compared to the rural environment of the Wisconsin campus. We have found that the ACH model of care can benefit many diverse patients, and the addition of the acquisition APP in this diverse urban environment flourished. More rural and isolated communities may need additional patient resources beyond the acquisition APP to significantly impact patient volumes.

We found that both of our ACH sites made advancements in acquisition efficiencies and workflows over the study period that drove down transition time. Although there was a trend towards decreased transition time at the acquisition APP site, this did not reach statistical significance. Although our site comparison did not show any significant decrease in patient ALOS while in the ED or BAM hospital, when looking at each geographic site independently, Wisconsin patients had a significantly longer ALOS in the ED or BAM hospital in the post-intervention timeframe, where the Florida site did not. This could indicate that although overall times waiting in the ED or BAM hospital prior to transition to ACH was increasing at both sites over time, the Florida site prevented a significant increase in this time by getting the option of ACH as an alternative to the physical hospital to these patients in a faster timeframe. Nevertheless, we believe that reducing the patients’ physical time in the ED or BAM hospital ward is crucial to both provide the best patient experience as well as drive down the cost of care which has been reported in other HaH studies [19,20].

One interesting finding in our study was that ALOS was not impacted by the addition of the acquisition APP. This may be due to the fact that we hypothesized that an acquisition APP would admit patients faster than with no acquisition APP, but as we found no statistical significance in transition time, this likely contributed to our ALOS outcomes. Interestingly, it may be that the transition time into ACH may not have any effect on ALOS. One possible reason for this is that the ACH is equivalent care to inpatient BAM care. Therefore, if a patient is being treated for a congestive heart failure exacerbation or an acute pneumonia, it does not matter on where the care is being delivered (BAM hospital vs. ACH) as the ALOS is linked to the diagnosis and treatment response, not to the patient location. This is important for two reasons.

First, there may be a fear amongst some hospital providers that the ALOS in HaH programs may be significantly longer than in BAM care as the patient is already located in the home, which removes the pressure to expedite discharge, especially in institutions with capacity issues. Past studies have shown that HaH programs often have a lower ALOS when compared to BAM hospital patients [21]. Therefore, the HaH model should be utilized without these fears of increased ALOS. ALOS reduction efforts may be better focused on diagnosis-direct quality improvement programs for both the BAM and HaH settings.

A second importance is that ALOS in BAM settings have been linked to costs [22,23]. Fear of prolonged ALOS in HaH and the subsequent costs associated may push some patients or providers to not use this modality. However, previous HaH studies have shown a reduction in comparative costs [19,20]. Our study shows that ALOS does not increase as we added an APP resource. A cost analysis should be conducted to see if the addition of the APP resources and the resultant increase in patient volumes into the program as a result (both improving program cost structure as well as opening BAM bed capacity) offset the cost of the APP FTE. Previous studies have shown that APPs in the BAM setting have been associated with high-quality patient care and reductions in ALOS [24,25]. Furthermore, APPs using telemedicine have been shown to manage highly complex chimeric antigen receptor (CAR) T-cell therapy in patients diagnosed with refractory diffuse large B-cell lymphoma in the outpatient setting [26]. Future studies focusing on the use of APPs in the HaH setting for patient acquisition, virtual patient assessment, and in-home care looking at care outcomes, ALOS, and patient experience are needed.

## 5. Limitations

Our study had several limitations. First, the sites differed in setting (urban Florida vs. rural Wisconsin) and resources (increased staff and supply chain resources in Florida vs. Wisconsin), which may have affected the outcomes. Second, as the ACH program was novel during the study period, other changes (building a patient identification registry in the EHR, educational lectures with ED physicians and hospitalists to increase ACH awareness, and improving ACH software (Cesia v2.1, Medically Home, Boston, MA, USA) for RN order entry and patient protocols) were ongoing to improve workflow, increase patient acquisition, and optimize care delivery, which may have affected the ALOS, transition time, and patient volume outcomes. Luckily, these efforts were being standardized through the ACH command center and should have been occurring in equal magnitude at both study sites. Additionally, the short time since the program’s initial launch and the small sample size decreases the power of this study. Moreover, the descriptive character of this analysis is associated with subjectivity in result interpretation and increases the risk of bias.

## 6. Conclusions

We found that the addition of a dedicated acquisition APP to our virtual hybrid hospital-at-home program resulted in a significant increase in patient volumes into the program without affecting ALOS or patient transition time when compared to our traditional admission process conducted by multiple providers on ACH team after completing patient rounds. Adding an acquisition APP may be an effective way that HaH programs can rapidly increase admission volumes, which are needed to scale HaH programs. These scaling efficiencies may help institutions with capacity constraints and ultimately reduce inpatient care costs. Future studies are necessary to see how to improve HaH throughput and patient transition times.

## Figures and Tables

**Figure 1 healthcare-11-00282-f001:**
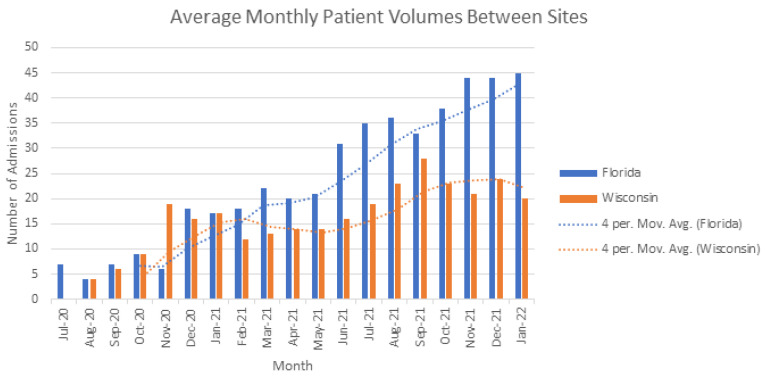
Average Monthly Patient Volumes Between Sites.

**Table 1 healthcare-11-00282-t001:** Patient Demographics.

Demographic Characteristics	Florida (n = 455)	Wisconsin (n = 298)	*p*-Value
Mean Age (SD)	70.2 (15.1)	70.2 (14.1)	*p* > 0.05 *
Sex n (%)			
Male	245 (53.8)	166 (55.7)	*p* > 0.05 ^†^
Female	210 (46.2)	132 (44.3)	
Race n (%)			
African American	29 (6.4)	0	*p* < 0.05 ^‡^
American Indian/Alaskan Native	2 (0.4)	0	
American born African	1 (0.2)	0	
Asian Filipino	12 (2.6)	2 (0.7)	
Choose not to disclose	7 (1.5)	2 (0.7)	
Other	12 (2.6)	0	
White	392 (86.2)	294 (98.7)	
Ethnicity n (%)			
Central American	0	2 (0.7)	*p* < 0.05 ^‡^
Choose not to Disclose	11 (2.4)	6 (2.0)	
Hispanic or Latino	11 (2.4)	2 (0.7)	
Not Hispanic or Latino	426 (93.6)	288 (96.6)	
Other Spanish Culture	3 (0.7)	0	
Puerto Rican	4 (0.9)	0	

Data reported as mean (± standard deviation) for age and as N (%) for others. SD: Standard deviation. * *p* values were calculated using two sample *t*-test. ^†^
*p* values were calculated using the Pearson’s Chi-squared test. ^‡^
*p* values were calculated using the Fisher’s Exact test.

**Table 2 healthcare-11-00282-t002:** Comparison of ALOS and Patient Volumes between Florida and Wisconsin Pre and Post Florida Intervention.

	Florida	Wisconsin	95% Confidence Interval ^†^	*p*-Value ^†^
PRIOR TO ACQUISITION APP (6 July 2020–31 May 2021)				
ED and/or BAM hospital LOS (mean days)	2.40	2.00	[−0.18, 0.98]	0.17
ACH Virtual Inpatient Care LOS (mean days)	3.72	3.87	[−0.84, 0.53]	0.65
Total Combined Physical and Virtual Hospital LOS (mean days)	6.12	5.87	[−0.84, 1.34]	0.64
Transition Time from ACH Consult to Home Admission (mean days)	1.32	1.87	[−1.21, 0.11]	0.10
Average number of patients acquired by the ACH program each month (n)	13.6	12.4	[−4.33, 6.62]	0.67
AFTER ACQUISITION APP INITIATION (1 June 2021–31 January 2022)				
ED and/or BAM hospital LOS (mean days)	2.91	2.59	[−0.22, 0.87]	0.22
ACH Virtual Inpatient Care (mean days)	3.71	3.75	[−0.54, 0.47]	0.88
Total Combined Physical and Virtual Hospital LOS (mean days)	6.63	6.34	[−0.55, 1.12]	0.47
Transition Time from ACH Consult to Home Admission (mean days)	0.85	1.16	[−0.91, 0.28]	0.28
Average number of patients acquired by the ACH program each month (n)	38.3	21.6	[11.54, 21.46]	<0.01

Abbreviations: LOS: Length of Stay, ACH: Advanced Care at Home, APP: Advanced Practice Provider, ED: Emergency Department, BAM: Brick and Mortar. ^†^ Calculated by two-sample *t*-test assuming equal variances.

**Table 3 healthcare-11-00282-t003:** ALOS and Patient Volumes Pre and Post Florida Intervention based by Clinical Site.

	(6 July 2020–31 May 2021)	(1 June 2021–31 January 2022)	95% Confidence Interval ^†^	*p*-Value ^†^
FLORIDA				
ED and/or BAM hospital LOS (mean days)	2.40	2.91	[−1.19, 0.17]	0.13
ACH Virtual Inpatient Care LOS (mean days)	3.72	3.71	[−0.60, 0.61]	0.98
Total Combined Physical and Virtual Hospital LOS (mean days)	6.12	6.63	[−1.66, 0.64]	0.37
Transition Time from ACH Consult to Home Admission (mean days)	1.32	0.85	[−0.07, 1.00]	0.08
Average number of patients acquired by the ACH program each month (n)	13.6	38.3	[−30.91, −18.49]	<0.01
WISCONSIN				
ED and/or BAM hospital LOS (mean days)	2.00	2.59	[−1.04, −0.13]	0.01
ACH Acute Phase LOS (mean days)	3.87	3.75	[−0.56, 0.80]	0.71
Total Combined Physical and Virtual Hospital LOS (mean days)	5.87	6.34	[−1.34, 0.40]	0.27
Transition Time from ACH Consult to Home Admission (mean days)	1.87	1.16	[−0.06, 1.47]	0.07
Average number of patients acquired by the ACH program each month (n)	12.4	21.8	[−13.69, −5.01]	<0.01

Abbreviations: LOS: Length of Stay, ACH: Advanced Care at Home, APP: Advanced Practice Provider, ED: Emergency Department, BAM: Brick and Mortar. ^†^ Calculated by two-sample *t*-test assuming equal variances.

## Data Availability

All patient data are held on Mayo Clinic research servers and access to data is restricted for patient privacy. However, if deemed necessary, data will be provided by the corresponding author upon reasonable request after approval from the needed institutional committee.

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
