# Peer review of "Impact of an Acquisition Advanced Practice Provider on Home Hospital Patient Volumes and Length of Stay"

_healthcare, 2023, doi:10.3390/healthcare11030282_

Round 1

Reviewer 1 Report

Dear Authors

The paper addressed an important topic, but presentation of results should be improved, for example reporting some figures of the main results

Author Response

  1. The paper addressed an important topic, but presentation of results should be improved, for example reporting some figures of the main results

--Thank you. We have added a figure (line 261) that shows our average patient volume growth by site and the moving average that widens after the intervention.

Reviewer 2 Report

This is a well-conducted study on Impact of a Patient Acquisition Advanced Practice Provider on Patient Volumes and Length of Stay in a Hospital-at-Home Program.   However, there are some comments that should be addressed. These are noted as given below.

I strongly suggest that the title of the research paper needs to be changed as accurate and meaningful. I have given some details in changing title.

1.    The  papers receive fewer citations if the title of the paper contains question marks, hyphens or colons

2.    Make sure the title is between 5 and 15 words in length

3.    A study found (http://mr.crossref.org/iPage?doi=10.6061%2Fclinics%2F2012%2805%2917) that papers get more views and citations if the title contains fewer than 95 characters.

Comments:

1.    Could authors provide the evidence for their argument of “Power analysis and sample size calculation were not conducted as this study was a retrospective review over a fixed time period

2.    In Table 1,

·       For age, authors need to present as Mean (SD), age in years and then 70.2(15.1) for Florida and 70.2(14.1) for Wisconsin (no need ± sign before SD)

·       For sex and other covariates,

Sex ,n(%) and then

For male – 245(53.8) for Florida and 166(55.7) for Wisconsin (no need % within parenthesis.

Race, n(%) and so on.

3.    In Table 2 and 3, why authors presented SD with confidence interval which is unusual way to represent the data. Authors have to present as effect size (confidence interval)

Effect size may be mean, OR, IRR, HR, etc. SD is not an effect size.

I am not sure that what is the purpose of these two Tables (2 &3). Can authors provide some information regarding the importance of these Tables.

Author Response

  1. I strongly suggest that the title of the research paper needs to be changed asaccurate and  I have given some details in changing title. The papers receive fewer citations if the title of the paper contains question marks, hyphens or colons. Make sure the title is between 5 and 15 words in length. A study found (ref) that papers get more views and citations if the title contains fewer than 95 characters.

--Thank you for this input; it was very interesting to read the reference as well. We have shortened the title to 15 words / 88 characters and reworded to eliminate hyphens.

  1. Could authors provide the evidence for their argument of “Power analysis and sample size calculation were not conductedas this study was a retrospective review over a fixed time period”

--Thank you for pointing this out. We have used a post-hoc t-test statistical power formula to test the power of our intervention population. We have added this to line 163-165 (old text removed) in the methods section and the power calculation results to line 177-178 of the results.

  1. In Table 1,For age, authors need to present as Mean (SD), age in years and then 70.2(15.1) for Florida and 70.2(14.1) for Wisconsin (no need ±sign before SD). For sex and other covariates,Sex, n(%) and then for male – 245(53.8) for Florida and 166(55.7) for Wisconsin (no need % within parenthesis.Race, n(%) and so on.

--We have made the suggested changes to table 1

  1. In Table 2 and 3, why authors presented SD with confidence interval which is unusual way to represent the data. Authors have to present as effect size (confidence interval) Effect size may be mean, OR, IRR, HR, etc. SD is not an effect size.

--Thank you. We have reformatted both tables to display the data as mean days or volumes each site followed by 95% confidence intervals and then p-values.

  1. I am not sure that what is the purpose of these two Tables (2 &3). Can authors provide some information regarding the importance of these Tables.

--Table 2 summarizes the data between the 2 sites for comparison in order to organize the results. Table 3 looks at each site independently and compares the before and after intervention times at each site. We have expanded the explanation of table 3 (lines 198-202) to better describe why we added it. Basically, although our intervention was the addition of the patient acquisition APP, other interventions were done in unison at both sites for program improvement (meaning each ACH site updated its workflows, staffing, transportation resources, and automation). These improvements done in unison at both sites helped each site become more efficient and grow volumes, but we wanted to show that the Florida site truly did grow its volumes significantly more. So it was just an extra measure to help dive this point home; not absolutely necessary, but we thought it would help answer that question for readers if they pondered it.

Reviewer 3 Report

Abstract

Line 13 – patients who required any type of care, regardless of the level of care?

Introduction

Line 35 - I suggest describing which positive aspects can occur for the patient in the reduction of the length of stay, in addition to the reduction of costs. I also suggest updating the reference cited from 2015 regarding hospital costs per patient.

Line 41 – I suggest focusing on the positive aspects of the telemedicine program for the patient and not just for the service.

Line 58 – I suggest reviewing the writing

Materials and methods

Line 63 – I suggest describing the difference between the rural and urban environment at the time of describing clinical setting, as these characteristics were mentioned in the study limitations.

Could patients be recruited from any clinic in the hospitals? If so, I suggest making it clear to the reader. How many professionals participated in the study? Did these professionals work directly in the services or perform tasks related only to the development and application of research phases and interventions?

Discussion

- According to the results, the percentages of race and ethnicity were significant in the analysis. Considering aspects related to the characteristics of adherence to the study, regional, sociodemographic and population aspects of the country, such as access to health insurance, what is the impact of these results on society? I suggest reflecting.

- Has any other study developed protocols similar to the research? If so, what are the main results?

Limitations

Line 278 – What other changes? I suggest quoting

Author Response

Abstract

  1. Line 13 – patients who required any type of care, regardless of the level of care?

--Thank you. We have clarified “in clinically stable medical patients” in this line

Introduction

  1. Line 35 - I suggest describing which positive aspects can occur for the patient in the reduction of the length of stay, in addition to the reduction of costs. I also suggest updating the reference cited from 2015 regarding hospital costs per patient.

--Thank you. We have added a line about the risks of prolonged LOS (infection, VTE, cognitive impairment) and linked the goal of reducing LOS to reduce these as well as costs.

  1. Line 41 – I suggest focusing on the positive aspects of the telemedicine program for the patient and not just for the service.

--We have added positive aspects of telemedicine including connecting to isolated/rural communities, positive patient experience, and positive provider experience.

  1. Line 58 – I suggest reviewing the writing

--Thank you. We have rewritten our hypothesis to run much smoother and be clearer to readers (now lines 64-68)

Materials and methods

  1. Line 63 – I suggest describing the difference between the rural and urban environment at the time of describing clinical setting, as these characteristics were mentioned in the study limitations.

--Thank you. We have added to this line (now line 78) descriptions of the urban nature of the Florida campus vs the rural nature of the Wisconsin campus, along with population density for each area from the US census.

  1. Could patients be recruited from any clinic in the hospitals? If so, I suggest making it clear to the reader. How many professionals participated in the study? Did these professionals work directly in the services or perform tasks related only to the development and application of research phases and interventions?

--We clarified in line 97-99 that patients were only transferred into ACH from the ED or inpatient wards (this is a CMS waiver requirement at this time for Medicare patients, that makes up the majority of our population). The work on this project was done by the primary ACH team which is the clinical team, made up of physicians, RNs, APPs, and our partner administrators that assist us in running daily ACH operations.

Discussion

  1. According to the results, the percentages of race and ethnicity were significant in the analysis. Considering aspects related to the characteristics of adherence to the study, regional, sociodemographic and population aspects of the country, such as access to health insurance, what is the impact of these results on society? I suggest reflecting.

--Very good point. In lines 252-259 we have added this difference between the sites to the results and reflected on the likely “why” (urban vs rural) , the positives (good that diverse group got more ACH care), and what this means for rural health ( patient isolation, may need more resources to be successful.

  1. Has any other study developed protocols similar to the research? If so, what are the main results?

--As far as we know, no other hospital at home program has studies this protocol change.

Limitations

  1. Line 278 – What other changes? I suggest quoting

--Thank you. We added a line (316-318) going over several key changes (building a patient identification registry in the EHR, educational lectures with ED physicians and hospitalists to increase ACH awareness, and improving ACH software for RN order entry and patient protocols) in order to clarify.